# Constraint-Based Spectral Space Template Deformation for Ear Scans

Srinivasan Ramachandran *
École de technologie supérieure, Montréal, Canada

Tiberiu Popa[†]
Concordia University, Montréal Canada

Eric Paquette[‡]
École de technologie supérieure, Montréal, Canada

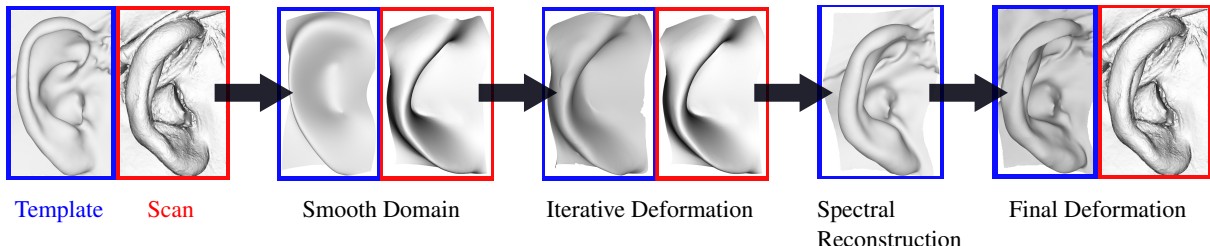

Template    Scan      Smooth Domain     Iterative Deformation     Spectral Reconstruction     Final Deformation

Figure 1: Our approach deforms a template to match the shape of a scan by aligning and deforming in a smooth domain space.

## ABSTRACT

Ears are complicated shapes and contain a lot of folds. It is difficult to correctly deform an ear template to achieve the same shape as a scan, while avoiding the reconstruction of noise from the scan and being robust to bad geometry found in the scan. We leverage the smoothness of the spectral space to help in the alignment of the semantic features of the ears. Edges detected in image space are used to identify relevant features from the ear that we align in the spectral representation by iteratively deforming the template ear. We then apply a novel reconstruction that preserves the deformation from the spectral space while reintroducing the original details. A final deformation based on constraints considering surface position and orientation deforms the template ear to match the shape of the scan. We tested our approach on many ear scans and observed that the resulting template shape provides a good compromise between complying with the shape of the scan and avoiding the reconstruction of the noise found in the scan. Furthermore, our approach was robust enough to scan meshes exhibiting typical bad geometry such as cracks and handles.

**Index Terms:** Computing methodologies—Computer graphics—Shape modeling—

## 1 INTRODUCTION

Virtual head models are extremely important in many fields ranging from game and entertainment industries to medical and cosmetic industries. Thus, new acquisition methods, striving to improve the accuracy and detail of 3D models, increase the level of capturing automation and decrease the overall cost and time of the acquisition. Despite the great efforts that have thus far been invested and the considerable research progress obtained, there are still areas that require exploration.

The first breakthrough in human head acquisitions was light stages [37] that could capture both the static geometry of the head as well as the appearance model. While the 3D reconstructions were impressive, due to the complexity and variability of the human face,

---

*e-mail: ashok.srinivasan2002@gmail.com

[†]e-mail: tiberiu.popa@concordia.ca

[‡]e-mail: eric.paquette@etsmtl.ca

it soon became evident that a one size fits all acquisition pipeline is not the best approach and many special methods have been developed targeting the different components of the human head such as lips, jaws, teeth, eyes, etc.

One component that has received little attention are human ears. For example, ears are important to recreate a faithful avatar. They are also of particular importance for visual effects, video games, and virtual reality, where we aim for believable characters with specific visual traits. Acquisition of human ears is difficult as ear geometry comes with a richer anatomical structure with many components that exhibit complex folds. Moreover, the ear geometry exhibits a large degree of self-occlusion, which leads to missing geometry in the scans. Furthermore, hair is often found in the way, occluding the ear, which is also detrimental to the reconstruction of the fine details found in the ears. All these challenges make ear reconstruction difficult and many of the general purpose reconstruction methods available may yield undesirable artifacts.

There are two main steps to virtual head model acquisition: a geometric reconstruction step where an unstructured scan is obtained, and a registration step where a template is deformed to match the shape of a scan. Typical registration methods first compute a set of 3D feature matches between the template and the scan followed by a deformation step where the template is deformed to match the 3D features. Unfortunately human ears are pretty smooth and do not contain easily identifiable local features. Moreover, the variance of the human ear shape is high and the fold structure is complex.

In this work we propose a non-rigid template registration method tailored to the specific geometric characteristics of human ears such as the long folds exhibiting long ridges and valleys. Our approach uses an input that is noisy and a potentially incomplete irregular mesh of an ear, and it deforms a template to match the geometry of the ear. We frame the non-rigid registration as an optimization problem in a smooth domain where the ear geometry is simplified. Point correspondences are obtained using edge detection in the image space. Fine details of the scan are matched in a way that strikes a balance to avoid reconstructing noise while still reconstructing legitimate ear details. Our method preserves the semantic structure of the human ear and it is robust to the wide natural variations of the ear shape.

## 2 RELATED WORK

Digital human modeling has seen a great level of improvement towards human facial scanning [16, 37]. That level of quality came through complex capture setup and equipment. Scans from standard

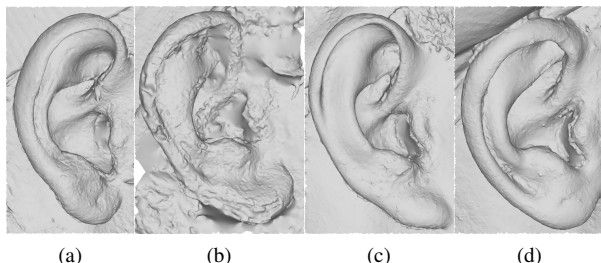

(a)      (b)      (c)      (d)

Figure 2: These ear scans demonstrate the diversity of ears with respect to their shape.

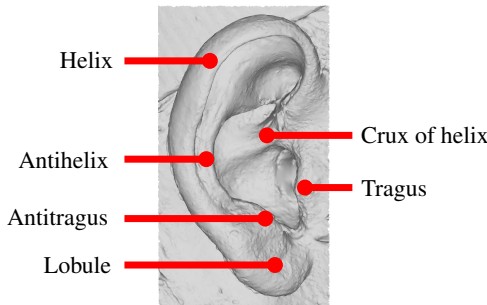

Figure 3: External ear anatomy is composed of many different components adding to the complexity of the ear shape.

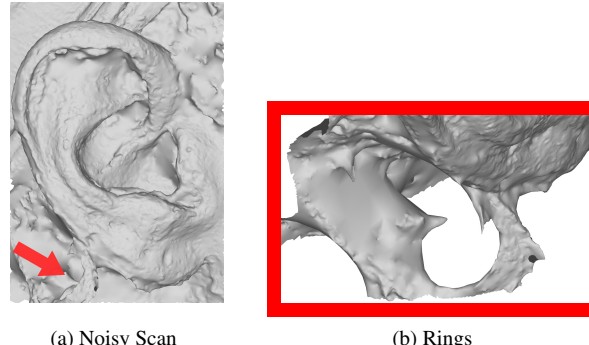

(a) Noisy Scan      (b) Rings

Figure 4: Ear scans often exhibit noise and have poor geometry, making them challenging for template matching. On top of noise, meshes often exhibit erroneous handles as seen in (b).

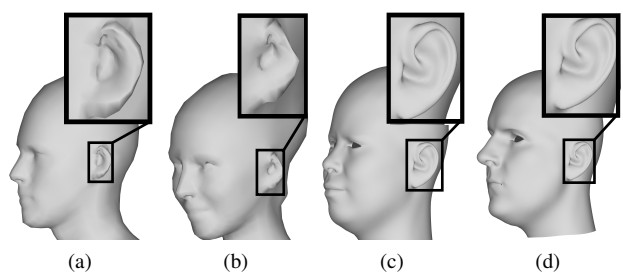

(a)      (b)      (c)      (d)

Figure 5: Example of scans from the Faust [13] (a)-(b) and Facewarehouse [14] (c)-(d) data sets. The ears in (a) and (b) show that standard template fitting methods perform poorly on ears. For the Facewarehouse examples in (c) and (d), while the geometry is smooth, it does not represent the ear anatomy fully. Details are missing from the anatomy such as the tragus, anti-tragus, anti helix. Furthermore, the ears are almost identical, probably because the template matching method "preferred" to reconstruct the scan to avoid picking up noise.

photogrammetry remain noisy and the noise is even harder to deal with when considering the intricate details found in the ear geometry. We will first present methods tailored to specific parts of the face. We will then review the methods specific to ears. We will end by discussing registration methods.

### 2.1 Facial Parts

General-purpose methods quickly showed their limits and researchers introduced methods specialized to specific parts of the face. Berard et al. focused on capture of eyes using a parametric model [10] and further improved on capture with a multi-view imaging system that can reconstruct poses of the eye [11]. Bermano et al. [12] present a method that works towards the reconstruction of eyelids including folding and wrinkles. For a realistic facial appearance it is important to consider the lips [19, 20] and jaw regions [42, 43]. For the teeth, it is important to capture their appearance and to fit them with respect to the mouth region [36, 38].

### 2.2 Ear Tailored Methods

Arbab-Zavar and Nixon [8] proposed a method that detects ears using elliptical Hough transforms. Ansari and Gupta [7] proposed a method that also detects ears in image space by detecting edges and segregating them into concave and convex edges, thus finding the outer helix region. Cummings et al. [17] detect ears in the images by modelling light rays and finding the helix regions. Other methods work in both image and depth space. Yan and Bower [39] proposed a method that detects the ears by combining both RGB and depth images. Chen and Bhanu [15] proposed a method that detects the helix region by analyzing discontinuities in the hills and valleys of a depth image. Zhou et al. [41] proposed a method that introduced histograms of categorized shapes for 3D ear recognition by adopting a sliding window approach and a support vector machine classifier. Ears are detected by analyzing the 3D features like saddles and ridges, and based on connectivity graphs in the depth images [32]. While the methods above are interesting, they are limited to ear

detection and do not provide solutions to the 3D ear reconstruction problem.

For the purpose of reconstruction, Guler et al. [6] proposed a method that computes a dense registration between an image and a 3D mesh template. It works based on convolution networks that learn a mapping from image pixels into a dense template grid. While the method is interesting, it considers images as inputs instead of 3D scans.

### 2.3 Dense Registration and Reconstruction in General

A dense registration can be a potential avenue in reconstructing a template to match the shape of a scan. Here we will focus on specific registration methods. For a more comprehensive list of dense registration methods the reader is referred to the survey of van Kaick et al. [35]. Ovsjanikov et al. [30, 31] proposed the functional maps framework to express dense registration. Lähner et al. [25] proposed a method that works by formulating the problem as matching between a set of descriptors, imposing a prior continuity on the mapping. A major limitation to this approach is that the difference in vertex density between meshes can be problematic. Furthermore, the choice of descriptors affects the results in the case of a noisy scan.

The Blended Intrinsic Maps (BIM) [24] method produces multiple low-dimensional maps that get blended in a global map. BIM suffers from distortion and discontinuities in its mappings. A ma-

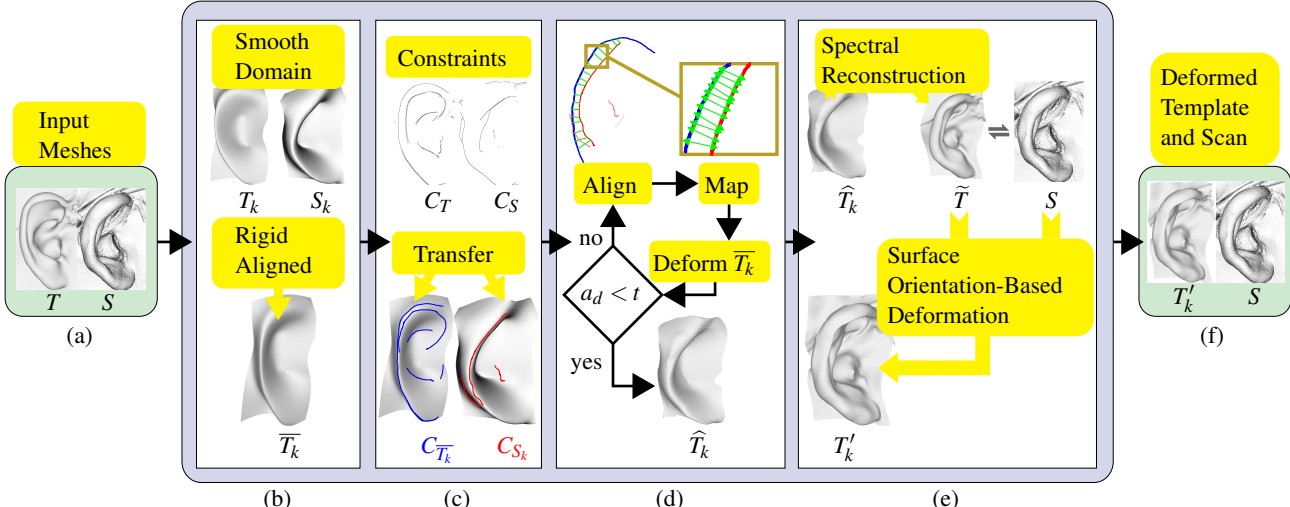

Figure 6: (a) Our inputs are a template and scan meshes $T$ and $S$. (b) $T$ and $S$ are transformed to a smooth domain and the template is rigid aligned to the scan. (c) Edges detected in the image space are transferred in 3D on the smooth meshes. (d) Smooth meshes are iteratively deformed through three sub steps of rigid alignment of constraints, constraints injective mapping, and non-rigid deformation of the smooth template. (e) Spectral reconstruction is used to reintroduce details to the template. This is followed by a last non-rigid deformation phase based on similar surface orientation constraints. (f) After these deformation phases, the template exhibits the shape of the scan without the noise and semantic regions are in correspondence.

jor limitation of this method is that it is difficult to be adapted for meshes with problems such as holes and noise.

Parametrization-based methods for dense registration work by transforming the meshes into a simpler space where finding a mapping between the meshes is easier. Athanasiadis et al. [9] proposed a geometrically-constrained optimization technique to map 3D genus-zero meshes on a sphere. Then, they morph the meshes with structural similarities by applying feature-based methods. Mocanu and Zaharia [29] proposed a two step spherical parameterization method 1) by analyzing the Gaussian curvature to align feature correspondences between the meshes, 2) by applying a morphing step to establish the mapping.

Another category of deformation methods work with user-given landmarks. Some methods [3, 4, 33] use the landmarks to cut the meshes, flatten them, and extract the dense registration from the planar domain or improve [22] an already provided registration in the planar domain. In addition to spherical and planar domains, other domains such as hyperbolic orbifolds have been proposed [1,2]. Landmark-based methods require a very carefully chosen and, in some cases, large set of corresponding landmarks.

Other methods deform the given meshes until their respective shapes match with each other; however, most of the methods are limited to near-isometric deformations [5]. Some methods [23, 26] overcome this limitation by trying to extend the range of objects to handle non-isometric pairs. However, these methods face practical challenges when dealing with scans that can contain noise and cracks in the mesh.

A practical limitation for most of the dense registration methods is the mesh quality of the scans. Moreover, a full-fledged dense registration also implies the picking up of noise from the scans thus leading to a bad reconstruction. One main goal of our approach is to achieve the conflicting goals of acquiring geometric detail while avoiding the reconstruction of the noise.

## 3 TEMPLATE DEFORMATION

An ear shape has a lot of variability as illustrated in Fig. 2, and its anatomy (Fig. 3) contributes to a shape that is complex in nature. Out of the box scanning methods struggle to reconstruct a good mesh

as can be seen in Fig. 2. Current methods work by either deforming a template or by dense registration, but they still fail on human ears largely owing to its complex anatomy and practical problems such as the mesh quality (Fig. 4). This is also exemplified in widely used data sets of faces. Fig. 5(a)-(b) shows heads from the Faust [13] data set. We can see that the mesh is of very poor quality in the ear region. Fig. 5(c)-(d) presents heads from the FaceWarehouse [14] data set. In this case, the ear geometry is good, but this is at the expense of not reconstructing a faithful ear (the ears are almost identical in the whole data set). These examples demonstrate that it is necessary to develop a novel approach specific for ears. Our approach fills the gap between methods picking up too much noise (Fig. 5(a)-(b)) and methods avoiding the noise at the expense of not reconstructing a faithful ear (Fig. 5(c)-(d)).

The inputs to our approach are two meshes uniformly scaled to fit in a unit cube: a template $T$ and a scan $S$ (Fig. 6(a)). Scan $S$ is a high-density mesh with holes, noise, and bad polygon quality (Fig. 4). We conduct a series of deformation phases to align the coarse and fine details of the template $T$ to the scan $S$. The input meshes are first converted to a smooth domain as $T_k$ and $S_k$. $T_k$ is rigidly aligned to $S_k$ and becomes $\overline{T_k}$ (Fig. 6(b)). Constraint points $C_T$ and $C_S$ are found on the input meshes $T$ and $S$ using edge detection. Constraint points $C_T$ and $C_S$ are transferred on $\overline{T_k}$ and $S_k$ as $C_{\overline{T_k}}$ and $C_{S_k}$ (Fig. 6(c)). $\overline{T_k}$ is iteratively deformed to match the shape of $S_k$ by aligning the constraints $C_{\overline{T_k}}$ with $C_{S_k}$ (Fig. 6(d)). At the end of the iterations, $\overline{T_k}$ becomes $\widehat{T_k}$ that approximates the coarse shape of $S_k$. With a spectral reconstruction $\widehat{T_k}$ is deformed as $\widetilde{T_k}$ that reintroduces the fine details lost in the smooth domain, but preserves the deformation undergone in the smooth domain. A closest location approach between similarly oriented areas results in new constraints used for the final deformation of the template to match the scan (Fig. 6(e)).

### 3.1 Smooth Domain Transformation

In this section we explain our first deformation phase where we align the coarse features of the ear (Fig. 6(b)). This phase uses spectral processing to transform our template and scan meshes into

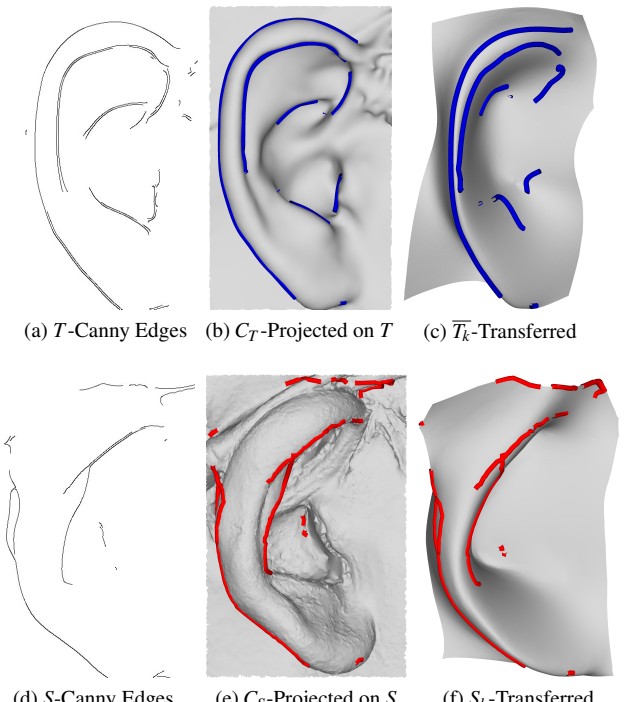

(a) *T*-Canny Edges  (b) $C_T$-Projected on *T*  (c) $\overline{T_k}$-Transferred

(d) *S*-Canny Edges  (e) $C_S$-Projected on *S*  (f) $S_k$-Transferred

Figure 7: Canny edge detection in image space (a, d) and its projection (b, e) on the meshes *T* and *S*. The edges are then transferred to $\overline{T_k}$ and $S_k$ (c, f). The canny edge detection is effective in identifying the sematic regions of the ears such as the helix, tragus, anti-tragus, etc.

a smooth domain where non-rigid registration is easier to perform. This spectral processing takes advantage of the eigenvectors of the Laplacian matrix of the mesh. Given the $n \times 3$ matrix of vertex positions, we compute the positions in the smooth domain as follows:

$$V' = U_k \cdot U_k^\mathsf{T} \cdot V, \qquad (1)$$

where $V'$ are the resulting vertex positions of the eigensubspace projection and $U_k$ is a $n \times k$ matrix containing the first $k$ eigenvectors. It can be noticed that with a full eigendecomposition, i.e., $k = n$, $U_{k=n} \cdot U_{k=n}^\mathsf{T}$ results in an identity matrix. By reducing $k$, $U_{k<n} \cdot U_{k<n}^\mathsf{T}$ removes less important details, but maintains the global shape of the mesh. For the purpose of eigendecomposition, the Laplacian matrix based on the cotangent weights [28] is used. Applying this transformation to *T* and *S* using their respective eigenvectors $U_k(T)$ and $U_k(S)$, they become $T_k$ and $S_k$. Mesh $T_k$ is then rigid-aligned [40] with $S_k$ and is now $\overline{T_k}$.

### 3.2 Ear Features and Constraints

This section describes how we extract meaningful features of the ears that we will use as constraints for the deformation in the next section. Our constraints are automatically computed using Canny edge detection on renderings of the 3D ears. An edge detection on 2D renderings of the original ears (*S* and *T*) proved to be quite robust in systematically detecting meaningful edges. To achieve this, an orthographic camera view was used for the rendering where the camera is facing the ears such that the view covers the bounding box of the ears. Each rendered image is 600px in height and the width varies between 400px to 500px based on the width of the ear. As it can be seen in Fig. 7(a) and (d), the edge detection identifies important semantic features of the ears. The detected edges (2D

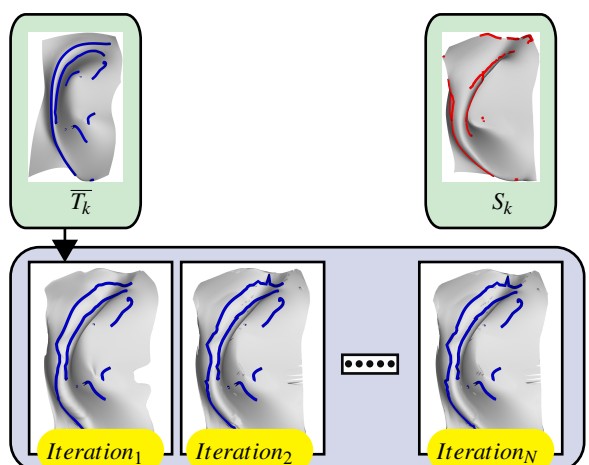

Figure 8: The figure shows how $\overline{T_k}$ deforms in a non-rigid fashion based on its constraints $C_{\overline{T_k}}$. As the iterations progress the movement becomes marginal and hence the iterations are terminated when the average of vertex movements reaches a threshold.

pixel locations) are then projected back to the mesh (Fig. 7(b) and (e)). These constraints $C_T$ and $C_S$ are 3D locations on the surface *T* and *S* respectively. The constraints are then transferred to $\overline{T_k}$ and $S_k$ (Fig. 7(c) and (f)) as $C_{\overline{T_k}}$ and $C_{S_k}$. The transfer relies on barycentric coordinates on the triangles of $T/\overline{T_k}$ and $S/S_k$.

### 3.3 Iterative Coarse-Level Deformation

This deformation phase is done by iterations consisting of two sub-steps: 1) constraints alignment and 2) non-rigid deformation. These iterations deform the smooth domain mesh $\overline{T_k}$ to match the shape of $S_k$ using the constraints $C_{\overline{T_k}}$ and $C_{S_k}$. A rigid alignment is applied from $C_{\overline{T_k}}$ to $C_{S_k}$. After rigid alignment, for each of the constraints in $C_{\overline{T_k}}$ a closest correspondence in $C_{S_{k_i}}$ is found. $\overline{T_k}$ is then deformed to align the constraints $C_{\overline{T_k}}$ on corresponding constraints from $C_{S_k}$. Each of the following iterations uses the updated $\overline{T_k}$ and $C_{\overline{T_k}}$. As the iterations progress the mesh $\overline{T_k}$ is deformed to match the shape of $S_k$.

**Constraints Alignment and Mapping** The first step of the iteration is the alignment of constraints $C_{\overline{T_k}}$ to $C_{S_k}$ using Go-ICP [40]. An injective map is found between the two sets of constraints $C_{\overline{T_k}} \to C_{S_k}$ based on closest locations.

**Non-Rigid Deformation** In the second step, the mesh $\overline{T_k}$ is deformed in a non-rigid fashion through an energy minimization composed of two terms. One term maintains the shape through Laplacian surface editing (LSE) [27], while the other term minimizes the distance between the constraints. Both terms are equally weighted with a value of 1.0.

The iterations end when the average movement of the constraints in one iteration, $a_d$, is within a threshold $t$ (Fig. 8). The examples shown in this paper rely on a threshold of $t = 10^{-6}$ (we can rely on an absolute threshold as we unitize the input meshes). The final version of $\overline{T_k}$ is referred to as $\widehat{T_k}$.

### 3.4 Spectral Reconstruction

Once the iterations are finished, $\widehat{T_k}$ will look similar (at a coarse level) to $S_k$. The template fitting is improved at the fine level with a reconstruction process that reintegrates the surface details while preserving the deformation undergone in the smooth domain. Reintroducing details and using surface orientation to create deformation

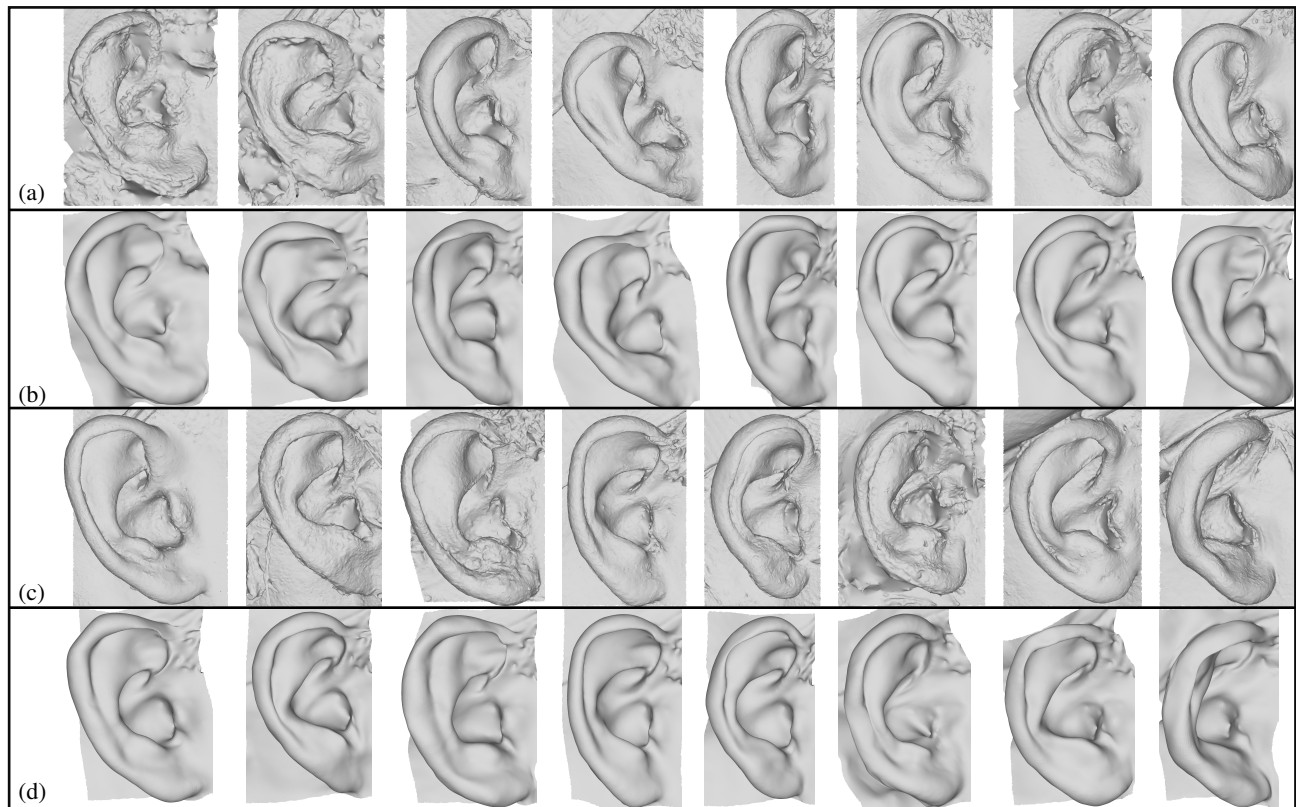

Figure 9: Results showing the deformation of the template ear mesh for 16 scans. Rows (a) and (c) are the original high-density ear scans of real people. Rows (b) and (d) are the results of template deformation using the pipeline of the presented approach.

constraints are important since unrelated mesh features can overlap in the smooth domain. Our spectral reconstruction is an important contribution of the presented approach. The idea is to express the deformation made in the smooth domain back to the original space. A similar idea has been proposed by Dey et al. [18]. In their method, they calculate a displacement vector between the vertex position of the original shape and the smooth domain. They then add this displacement vector back to the deformed smooth domain. Instead of using local displacement vectors, our approach strives for a smooth surface by globally enforcing the original surface Laplacians while preserving the smooth domain deformation.

In our approach we reconstruct using the Laplacians of $T$ combined with the spectral coefficients from $\widehat{T}_k$. The goal is to reconstruct a surface that maintains the details of mesh $T$ while keeping the transformations of $\widehat{T}_k$. This is done by solving for vertex positions under two constraints. The first constraint tries to maintain the Laplacian coordinates of the original mesh $T$:

$$L(T)V(\widetilde{T}) = L(T)V(T), \qquad (2)$$

where $V(\widetilde{T})$ is a $n \times 3$ matrix representing the vertices after reconstruction. For the second constraint, we want to maintain the transformation undergone in the smooth domain. To do so, we work in the low dimensional space of the eigenvectors $U_k(T)$. In that space, we try to maintain the same coordinates as those of $\widehat{T}_k$:

$$U_k(T)^\mathsf{T}V(\widetilde{T}) = U_k(T)^\mathsf{T}V(\widehat{T}_k), \qquad (3)$$

where $V(\widehat{T}_k)$ are the vertex coordinates of $\widehat{T}_k$.

We then solve for the vertex positions $V(\widetilde{T})$ that meet these two constraints in a least-squares sense:

$$\underset{V(\widetilde{T})}{\arg\min} \sum_{i=1}^{n} \left\| L_i(T)V_i(\widetilde{T}) - L(T)V(T) \right\|_2^2 +$$
$$\sum_{i=1}^{n} \left\| U_k^i(T)^\mathsf{T}V_i(\widetilde{T}) - U_k^i(T)^\mathsf{T}V_i(\widehat{T}_k)) \right\|_2^2 .V \quad (4)$$

Solving Equation 4 applies the changes made in the smooth domain back to the original domain thereby deforming $\widehat{T}_k$ to $\widetilde{T}$, a shape that is similar to $S$ at the coarse level.

Our spectral reconstruction strategy (Equation 4) is composed of two parts. The LSE alone through the $L(T)$ matrix is rank deficient resulting in an underconstrained system. Our second term involving $U_k$ contains $k$ additional constraints. The eigenvectors of $U_k$ are orthogonal to each other and the resulting system can be solved in a stable way.

### 3.5 Fine-Level Constraint Based Deformation

Given the previous deformation steps, the shape of $\widetilde{T}$ is similar to $S$, but only at a coarse level. By exploiting the spectrally reconstructed surface with more details, the template is further deformed with positional constraints based on similar surface orientation. The goal is to deform the template to match the shape of the scan without inheriting the surface noise from the scan. Hence, we use LSE to deform $\widetilde{T}$ while maintaining a smooth surface. We identify constraints as closest locations on the surface of $S$ for each vertex of $\widetilde{T}$. We reject some of the closest location correspondences based on surface orientation. We keep only those correspondences that have an angle between the vertex normal on $\widetilde{T}$ and the normal at the closest location on $S$ is less than a threshold (we used $d = 45°$). The selected

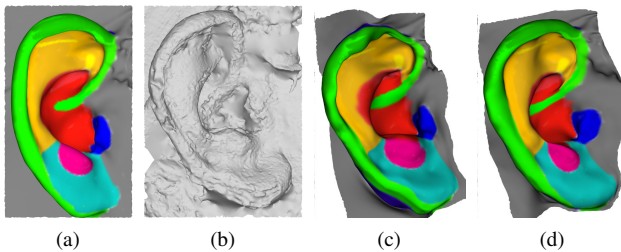

| (a) | (b) | (c) | (d) |

Figure 10: (a) Template with its features colored. (b) A scan from our data set. (c) Deformation using mapping without reconstruction from spectral shape. The semantic correspondence is wrong at various places that can be identified on areas showing incorrect lateral sliding and bad reconstruction. (d) Deformation after spectral reconstruction. Both the surface quality and the semantic region correspondences greatly improved.

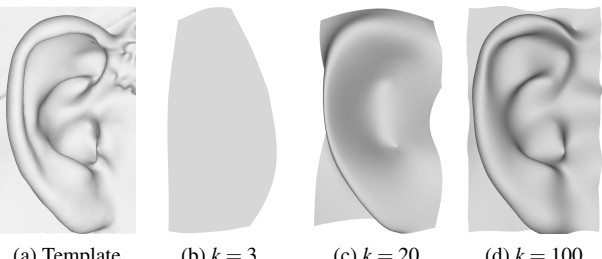

| (a) Template | (b) $k = 3$ | (c) $k = 20$ | (d) $k = 100$ |

Figure 11: Different selection of $k$ and the resulting shapes. (a) The original template ear. (b) Smooth domain transform using $k = 3$ results in a planar shape that is too simple to allow the alignment of any features. (d) Smooth domain transform using $k = 100$ results in a more detailed shape, but it contains a lot of folds that hinder the alignment. (c) A value of $k = 20$ results in a shape that is simple but detailed around the helix region, which is an important feature we want to align in the smooth domain.

locations are used as anchor locations for the LSE deformation. As in Sect. 3.3, both terms are equally weighted with a value of 1.0. This deforms $\widetilde{T}_k$ to the final template shape.

## 4 RESULTS AND DISCUSSION

The scans in our data set were acquired using a multi-view stereo setup. Each scan consists of a dense polygonal mesh that has between 90$k$ to 120$k$ vertices. Note that we plan to release a subset of our data set for other researchers. Fig. 9 shows the scans (a and c) and the deformed template (b and d) using the presented pipeline. The scans exhibit a lot of diversity in their shape and hence the deformation of the template to match the shape was quite a challenge. These meshes are irregular, can have erroneous or missing geometry, degenerate triangles as well as topological errors as illustrated in Fig. 4. The helix region of the scans were the worst affected during the capture due to the presence of interfering hair strands. We can see that our approach is robust against noise while being able to match the shape of the ears.

Our implementation uses OpenCV for edge detection in image space, Python packages SciPy and NumPy for linear system solutions, and Blender for mesh handling and the LSE system.

### 4.1 Spectral Reconstruction Benefits

We tested our approach with and without the spectral reconstruction. Without spectral reconstruction, we apply the deformation of Sect. 3.5 directly on $\widehat{T}_k$, skipping the reconstruction phase of Sect. 3.4. While the iterative deformation finishes with $\widehat{T}_k$ that exhibits a shape globally similar to $S_k$, the lack of detail is problematic for the deformation of the template to match $S$. Fig. 10 shows a scan and the template with semantically important features highlighted with colors. Fig. 10(c) shows a typical example of incorrect lateral sliding when the final deformation is done directly from $\widehat{T}$. When the final deformation is conducted from $\widetilde{T}$ to $S$ based on similarly oriented locations, we observe a significant improvement in fidelity of the shape as well as the correspondence of semantic regions (Fig. 10(d)). Furthermore, our spectral reconstruction approach is general. For example, like the method of Dey et al. [18], our approach could be applied to animation.

### 4.2 Selection of $k$ Eigenvectors

The selection of $k$ for the eigen decomposition is crucial as it plays a major role in the pipeline. The idea behind the selection of $k$ impacts a) how simple the shape will be in the smooth domain (Sect. 3.1) and b) the number of constraints available for spectral reconstruction (Sect. 3.4). Fig. 11(b) shows a smaller selection of $k = 3$ that

results in a simpler shape that does not contain any features, preventing the alignment of coarse features in the smooth domain. It also results in a very small number of constraints for spectral reconstruction. Fig. 11(d) shows a higher selection of $k = 100$ that results in higher number of constraints for spectral reconstruction, but the shape also reintroduces a lot of folds from the original surface (Fig. 11(a)). We observed that aligning both the coarse and fine details in a single step is very difficult. A selection of $k = 20$ (Fig. 11(c)) was balanced for both being a simpler shape with important coarse features, and also sufficient in terms of the number of constraints for spectral reconstruction.

### 4.3 Comparison to Mapping Method

Many mapping methods require manual landmarks, and as such cannot be used for comparison with our fully automatic approach. Furthermore, many methods do not work with meshes such as our ear scans, which contain boundaries and bad geometry. Lähner et al. [25] propose a mapping method using SHOT descriptors [34]. Their approach fails with high density scans to produce dense mapping, however it is able to produce a sparse mapping of around 3000 vertex-to-vertex correspondences. We tested if we could use this sparse mapping to directly deform template $T$ to the shape of the scan, thus avoiding the use of the smooth domain. This sparse map was evaluated by deforming the template with the 3000 registrations points employed as constraints using LSE, an idea similar to the deformation phase explained in Sect. 3.5. The deformation using the constraints from Lähner et al. is compared with our final result $T'_k$. Fig. 12 shows the comparison of results for two different ears. From the results we can see that our approach from Sect. 3.5 performs better.

### 4.4 Limitations

Fig. 13 shows the worst results from our experiments. In most cases, the edge detection constraints (Sect. 3.2) and non-rigid deformation (Sect. 3.3) steps align the features of the ears quite well, but as can be seen in Fig. 13 the helix occasionally does not align completely. The region of the helix close to the crux of the helix is another region where the registered template and the scan are sometimes not in a very good correspondence.

## 5 CONCLUSION AND FUTURE WORK

We presented an approach to fit a template ear mesh to scans of real ears. The template and the scan are first transformed to an eigenspace smooth domain where we begin by conducting a rigid alignment

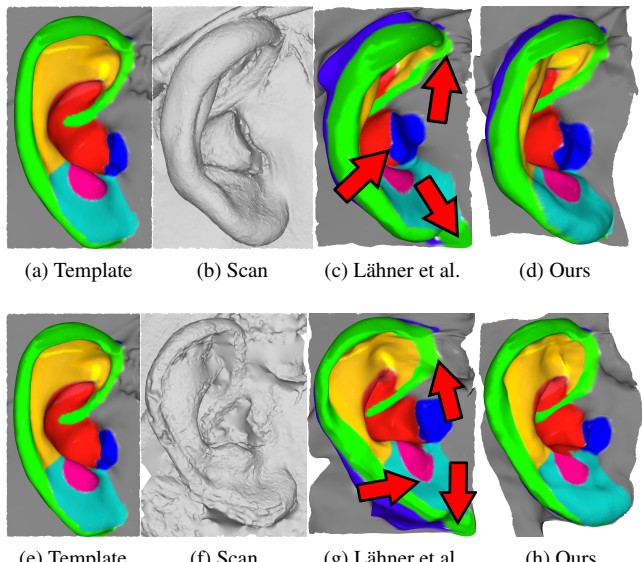

| (a) Template | (b) Scan | (c) Lähner et al. | (d) Ours |

| (e) Template | (f) Scan | (g) Lähner et al. | (h) Ours |

Figure 12: We compare (c) and (g) the results skipping the smooth domain by deforming the template directly using constraints derived from the method of Lähner et al. [25], to (d) and (h) using our pipeline involving the smooth domain. We can observe that the results skipping the smooth domain produce lateral sliding (indicated by red arrows) of the corresponding semantic regions.

on the smooth meshes. Features of the ears are detected in image-space with a Canny edge detection. The smooth domain eases the alignment of the coarse scale features of the meshes. The next phase iterates upon three sub steps of aligning the edge detection features, computing an injective mapping of the features from the template to the scan, and conducting a non-rigid deformation of the template through Laplacian surface editing with the features as constraints. We then reintroduce the details of the template mesh through a spectral reconstruction. Our spectral reconstruction optimizes for the spectral coordinates and for surface smoothness through Laplacian constraints. This generates a smooth surface with the details of the original template, while preserving the deformation from the smooth domain. The detailed template mesh is finally deformed through Laplacian constraints and constraints based on closest location of surface regions with similar orientation. One notable advantage of our approach is that it is robust against bad mesh quality and is also completely automatic. Moreover, we are convinced that our spectral reconstruction approach is general and could be used outside of the ear reconstruction pipeline. We will investigate this avenue as in future research work. The fixed template used in our approach could be replaced by a 3D Morphable Model (3DMM) of ears. In a similar fashion to the work of Donya et al. [21], we could adjust the 3DMM parameters to have a template ear that is already much closer to the geometry of the scanned ear. Finally, the current choice of constraints is limited to ears, but we believe that our series of deformation phases could be applied to other types of shapes. In this sense, deriving other types of constraints is an interesting direction for future work.

## 6   ACKNOWLDGEMENTS

This work was supported by Natural Sciences and Engineering Research Council of Canada (NSERC),[CRDPJ 535746 - 18]. We wish to thank Zorah Lähner from Technical University, Munich, Germany and author of Efficient Deformable Shape Correspondence via Kernel Matching [25] for sharing her code so that we could use

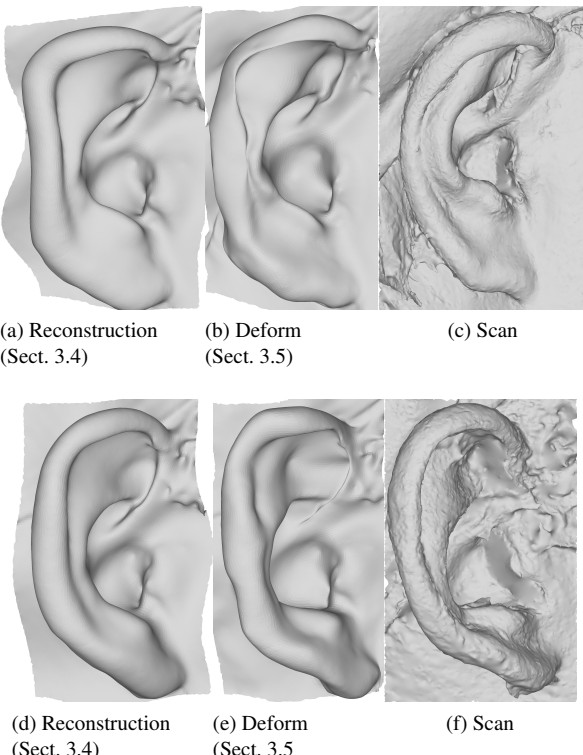

| (a) Reconstruction (Sect. 3.4) | (b) Deform (Sect. 3.5) | (c) Scan |

| (d) Reconstruction (Sect. 3.4) | (e) Deform (Sect. 3.5 | (f) Scan |

Figure 13: This figure presents our worst results (b) and (e). We can see the lack of alignment in the helix region. The alignment problem is apparent right at the spectral reconstruction step (a) and (d), Sect. 3.4) and originates from the edge detection constraints (Sect. 3.2) and the deformation in the smooth domain (Sect. 3.3).

it for comparison. We also want to thank the anonymous reviewers, and all the participants for the facial scanning.

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
