# OpenReview forum: "Constraint-Based Spectral Space Template Deformation for Ear Scans"
_graphicsinterface.org/Graphics_Interface/2020/Conference — GI 2020_

### Official Review · AnonReviewer2 · 2020-04-20
**Good Application of Spectral Space Registration to Ear Capture**

**Rating:** 7
**Confidence:** 4

**Review:**

The paper introduces a method to register ear scans. Capturing human geometry is an important topic as virtual humans and VR need accurate body reconstruction. This work improves the performance on registering ear scans to a template mesh by first non-rigidly registering a smoothed version of the template shape to the scanned shape, then adding back the details. The techniques (LSE, non-rigid ICP) used are not new, but the application to ear registration worths some merit.

One problem I find is: the helix root appears not to match very well on Figure 9, especially on the first example from the left on row (d). The authors do recognize the helix sometimes does not align completely. Maybe a note on the helix root can be added to give more details on limitations.

---

### Official Review · AnonReviewer3 · 2020-04-20
**Well written paper with good quality results**

**Rating:** 7
**Confidence:** 2

**Review:**

This paper presents a framework for reconstructing ear geometry from scan data by a series of steps to match and deform a template ear mesh.The paper appears quite comprehensive and the quality of the results is high.

I am not familiar with this area, so I cannot give a confident assessment. Nevertheless, I found the paper well written and was able to follow the exposition. I think the paper does a good job of showing the effect of each component of the system and communicates how it works as a whole. There are a few areas that could use a little further clarification, below:

"The first breakthrough in human head acquisitions was light stages that could capture both the static geometry of the head as well as the appearance model." - citation?

How do you pick the number of Eigenvectors k? It seems this would be quite an important factor?

Section 3.1, is there a citation for the method used to perform the rigid alignment?

Curious if there are alternatives for performing the edge detect phase in mesh-space rather than image space. I would have thought this would be simpler in some ways? Extending the discussion of the motivation for an image space method would be nice. How are constraints C_t,C_s represented? As barycentric coordinates of the triangle the image space points project onto?

For the non-rigid deformation, how are the two terms weighted? Does the user need to specify stiffness values, if so how are they set?

abstract: "while avoiding to reconstructing the noise" -> remove 'to'

"proposed a method that introduced a histograms" -> singular/plural mismatch

---

### Official Review · AnonReviewer1 · 2020-04-22
**Nice Paper with acceptable results**

**Rating:** 6
**Confidence:** 3

**Review:**

This paper assembles some well known concept like deformation transfer and spectral encoding of level of details. In essence this algorithm provides a top-down registration from a bad scan of ears into a template, allowing for a considerably better result. Templated fitting is not really a novel idea though I haven't seen this specific application around. Nevertheless, there is a plethora of data-driven registration methods for MRI (which is not a "geometry processing" problem per se but more of computer vision). As such, I think this paper is an interesting read for practitioners albeit not offering that much for researchers.

---

### Decision · Program_Chairs · 2020-04-25

Accept